# A SARS-CoV-2 neutralizing antibody with extensive Spike binding coverage and modified for optimal therapeutic outcomes

Yu Guo [1,15✉], Lisu Huang[2,15], Guangshun Zhang[1,3,15], Yanfeng Yao[4,15], He Zhou[5,15], Shu Shen [6,15], Bingqing Shen[5], Bo Li[1,3], Xin Li[1,3], Qian Zhang[5], Mingjie Chen[5], Da Chen[1,3], Jia Wu[5], Dan Fu[1], Xinxin Zeng[2], Mingfang Feng[5], Chunjiang Pi[5], Yuan Wang[1,3], Xingdong Zhou [1,3], Minmin Lu[5], Yarong Li[7], Yaohui Fang[6], Yun-Yueh Lu[5], Xue Hu[6], Shanshan Wang[5], Wanju Zhang[2], Ge Gao[4], Francisco Adrian[5], Qisheng Wang [8], Feng Yu[8], Yun Peng[4], Alexander G. Gabibov[9], Juan Min[4], Yuhui Wang[1,3], Heyu Huang[2], Alexey Stepanov [9], Wei Zhang [1,3], Yan Cai[10], Junwei Liu[10], Zhiming Yuan [4], Chen Zhang[1], Zhiyong Lou [11✉], Fei Deng [6✉], Hongkai Zhang [1,3,12,13✉], Chao Shan [6✉], Liang Schweizer [5✉], Kun Sun [2✉] & Zihe Rao [1,3,13,14✉]

COVID-19 pandemic caused by SARS-CoV-2 constitutes a global public health crisis with enormous economic consequences. Monoclonal antibodies against SARS-CoV-2 can provide an important treatment option to fight COVID-19, especially for the most vulnerable populations. In this work, potent antibodies binding to SARS-CoV-2 Spike protein were identified from COVID-19 convalescent patients. Among them, P4A1 interacts directly with and covers majority of the Receptor Binding Motif of the Spike Receptor-Binding Domain, shown by high-resolution complex structure analysis. We further demonstrate the binding and neutralizing activities of P4A1 against wild type and mutant Spike proteins or pseudoviruses. P4A1 was subsequently engineered to reduce the potential risk for Antibody-Dependent Enhancement of infection and to extend its half-life. The engineered antibody exhibits an optimized pharmacokinetic and safety profile, and it results in complete viral clearance in a rhesus monkey model of COVID-19 following a single injection. These data suggest its potential against SARS-CoV-2 related diseases.

[1] State Key Laboratory of Medicinal Chemical Biology and College of Pharmacy, Nankai University, Tianjin, People's Republic of China. [2] Xinhua Hospital, Shanghai Jiao Tong University School of Medicine, Shanghai, People's Republic of China. [3] College of Life Science, Nankai University, Tianjin, People's Republic of China. [4] Center for Biosafety Mega-Science, Wuhan Institute of Virology, Chinese Academy of Sciences, Wuhan, Hubei, People's Republic of China. [5] HiFiBio (Hong Kong) Limited, Central, Hong Kong. [6] State Key Laboratory of Virology and National Virus Resource Center, Wuhan Institute of Virology, Chinese Academy of Sciences, Wuhan, Hubei, People's Republic of China. [7] Wuxi Biologics Limited, Shanghai, People's Republic of China. [8] Shanghai Synchrotron Radiation Facility, Shanghai Advanced Research Institute, Chinese Academy of Sciences, Shanghai, People's Republic of China. [9] Shemyakin-Ovchinnikov Institute of Bioorganic Chemistry, Russian Academy of Sciences, Moscow, Russia. [10] Tianjin International Joint Academy of Biotechnology & Medicine, Tianjin, People's Republic of China. [11] MOE Key Laboratory of Protein Science & Collaborative Innovation Center of Biotherapy, School of Medicine, Tsinghua University, Beijing, People's Republic of China. [12] Shanghai Institute for Advanced Immunochemical Studies, ShanghaiTech University, Shanghai, People's Republic of China. [13] Frontiers Science Center for Cell Responses, Nankai University, Tianjin, People's Republic of China. [14] Guangzhou Laboratory, Guangzhou, Guangdong, People's Republic of China. [15]These authors contributed equally: Yu Guo, Lisu Huang, Guangshun Zhang, Yanfeng Yao, He Zhou, Shu Shen. ✉email: guoyu@nankai.edu.cn; louzy@mail.tsinghua.edu.cn; df@wh.iov.cn; hongkai@nankai.edu.cn; shanchao@wh.iov.cn; l.schweizer@hifibio.com; sunkun@xinhuamed.com.cn; raozh@mail.tsinghua.edu.cn

Coronavirus Disease 2019 (COVID-19) has recently emerged throughout the world as the largest pandemic of the twenty-first century, with more than 123 million confirmed cases and 2.7 million deaths worldwide as of March 22, 2021. A public health crisis on such a scale requires a range of effective prophylactic and treatment options. Neutralizing antibodies against the severe acute respiratory syndrome coronavirus 2 (SARS-CoV-2) Spike (S) protein become one of the promising options to treat and prevent COVID-19 pandemic, especially for the most vulnerable population including seniors, patients with underlying conditions such as immune deficiency, diabetes, cardiovascular, pulmonary, and kidney diseases. Like other Class I fusion proteins, the S glycoprotein forms homotrimers on the surface of the SARS-CoV-2 virus particle, mediates recognition of and binding to the human receptor, angiotensin-converting enzyme 2 (ACE2), through its receptor-binding domain (RBD), and then induces the virus–host cell membrane fusion. Therefore, S glycoprotein is regarded as the primary target for interfering with the virus entry process[1]. A number of SARS-CoV-2 vaccines currently under development or already in use have shown effectiveness in inducing anti-viral antibodies and preventing incidence and severity of COVID-19[2–5]. However, the success of SARS-CoV-2 vaccines ultimately hinges on the quality and longevity of the induced immune responses, particularly in elderly or individuals with pre-existing conditions, and on the acceptance of vaccination by the general public. Moreover, recent reports of SARS-CoV-2 reinfection[6,7] in patients pose a further challenge to the vaccine approach.

To date, significant efforts have been made to the discovery and development of SARS-CoV-2 neutralizing antibodies using various platforms, including antibody discovery from SARS-CoV[8,9] or SARS-CoV-2 convalescent patients[10–18], from humanized mice[19,20], and from phage libraries[21,22]. As a result of these efforts, several leading antibodies have entered the clinical stage with anti-viral efficacy demonstrated in mild-moderate COVID-19 patients[23]. Of note, a major concern for the development of these neutralizing antibody therapies is the potential risk of antibody-dependent enhancement (ADE) of infection[24,25], as previously reported in dengue[26] and SARS-CoV[27] infections. The risk of ADE was raised as a concern for SARS-CoV-2 countermeasures recently as well[18,28]. Another aspect to consider for an effective antibody therapy is the ability to target evolving mutated virus. By analyzing all reported sequences from GISAID (gisaid.org) from December 24, 2019 to December 13, 2020 with the LANL pipeline[29], mutations (with frequency ≥0.3% for non-ACE2 interface residues and ≥0.1% for ACE2 interface residues) to about 172 residues in the Spike protein have been identified.

Here in this work, we isolate and identify neutralizing antibodies binding to SARS-CoV-2 Spike protein from COVID-19 convalescent patients; one of the best RBD-specific antibodies, P4A1, shows potent neutralizing activities with nanomolar $IC_{50}$. The high-resolution complex structure analysis reveals its binding epitope, which covers majority of the binding site of hACE2. Importantly, P4A1 was subsequently engineered and results in an optimized pharmacokinetic (PK) and safety profile. The effectiveness of the engineered antibody P4A1–2A was further examined in a SARS-CoV-2 infection model in *rhesus macaques*, which demonstrate potent anti-viral efficacy following a single injection. Overall, these data suggest its potential for clinical development against SARS-CoV-2-related diseases.

## Results

**Antibody discovery and in vitro characterization.** To identify SARS-CoV-2 neutralizing antibodies from convalescent patients, we utilized the single-cell B-cell receptor sequencing approach. SARS-CoV-2 neutralizing antibodies were isolated from two convalescent patients with the highest titer against S protein (Patient 4 and Patient 20) among 23 patients studied (Table 1 and Supplementary Fig. 1a, b). SARS-CoV-2 S protein binding B cells were enriched from peripheral blood mononuclear cells (PBMCs) using biotinylated S protein-conjugated magnetic beads as described in Methods and Supplementary Fig. 1c, d. Among the sequenced antibodies, P4A1, P20A2, and P20A3 were shown to bind to the S1 subunit with sub-to-nM $EC_{50}$s, but they did not bind to the S2 subunit, while blocking the binding of the S1 subunit to ACE2-expressing cells with $IC_{50}$ values in the nM range (Fig. 1a). P4A1, P20A2, and P20A3 neutralized the live SARS-CoV-2 infection of Vero E6 cells with 50% neutralization dose ($ND_{50}$) values of 5.212, 43.79, and 29.03 nM, respectively (Fig. 1b). According to analysis based on the ImmunoGenetics database, the IGHV germline origin of antibody P4A1 belonged to IGHV3–53, which is one of the most frequent used germline families after SARS-CoV-2 infection in patients[17,30] (Supplementary Figs. 4 and 5a).

**Structural analysis of P4A1 Fab-RBD complex.** To gain insight into the structural basis for the blocking and neutralizing mechanism of P4A1, the structure of the antibody Fab/Spike-RBD complex was resolved by X-ray crystallography at the resolution of 2.1 Å (Fig. 2a and Supplementary Table 1). Three heavy-chain complementarity-determining regions (HCDRs), two light-chain complementarity-determining regions (LCDRs), together with weak interaction from the light-chain framework region 3 constitute the interaction network with SARS-CoV-2 RBD (Fig. 2b and Supplementary Table 2). The buried surface area (BSA) from the heavy-chain interaction in P4A1 (781 Å$^2$) is relatively larger compared to reported neutralizing antibodies including B38 (713 Å$^2$)[17], CB6 (734 Å$^2$)[18], CC12.1 (723 Å$^2$), and CC12.3 (698 Å$^2$)[16] (Supplementary Fig. 5). However, the BSA from the light chain varies significantly, with P4A1 (414.6 Å$^2$) compared to B38 (495 Å$^2$)[17], CB6 (334 Å$^2$)[18], CC12.1 (566 Å$^2$), and CC12.3 (176 Å$^2$)[16] (Supplementary Table 3). Specifically, the interaction network is mainly contributed by hydrogen bond and hydrophobic interaction. For the heavy chain, P4A1 HCDR1 and HCDR2 share similar interaction patterns with SARS-CoV-2 Spike RBD as CC12.1 and CC12.3, while these antibodies differ in their HCDR3 sequence, with Q100, E101 forming a hydrogen bond with K417 and Y453, and the side chain of L102 inserting into the hydrophobic cavity formed by L485, F456, and Y489 (Fig. 2c, upper panel). For the light chain, the hydrogen bond network is mediated by G28, S30, S31, W32 in LCDR1 with T500, N501, G502, Q498, Y449, and G496; and E90, N92, S93 in LCDR3 with Y505, and R403 (Fig. 2c, lower panel). It is worth noting that the water molecule plays a more important role for the light chain than it does for the heavy chain.

**Table 1 Patient sample information for FACS.**

| | Patient no. | Patient gender | Cell number | Cell viability (%) |
|---|---|---|---|---|
| PBMC | Patient 4 | Female | $2.54 \times 10^5$/mL, 200 μL | 84 |
| PBMC | Patient 20 | Male | $7.1 \times 10^5$/mL, 200 μL | 90 |
| PBMC | Healthy donor | Male | $7.0 \times 10^5$/mL, 200 μL | 92 |

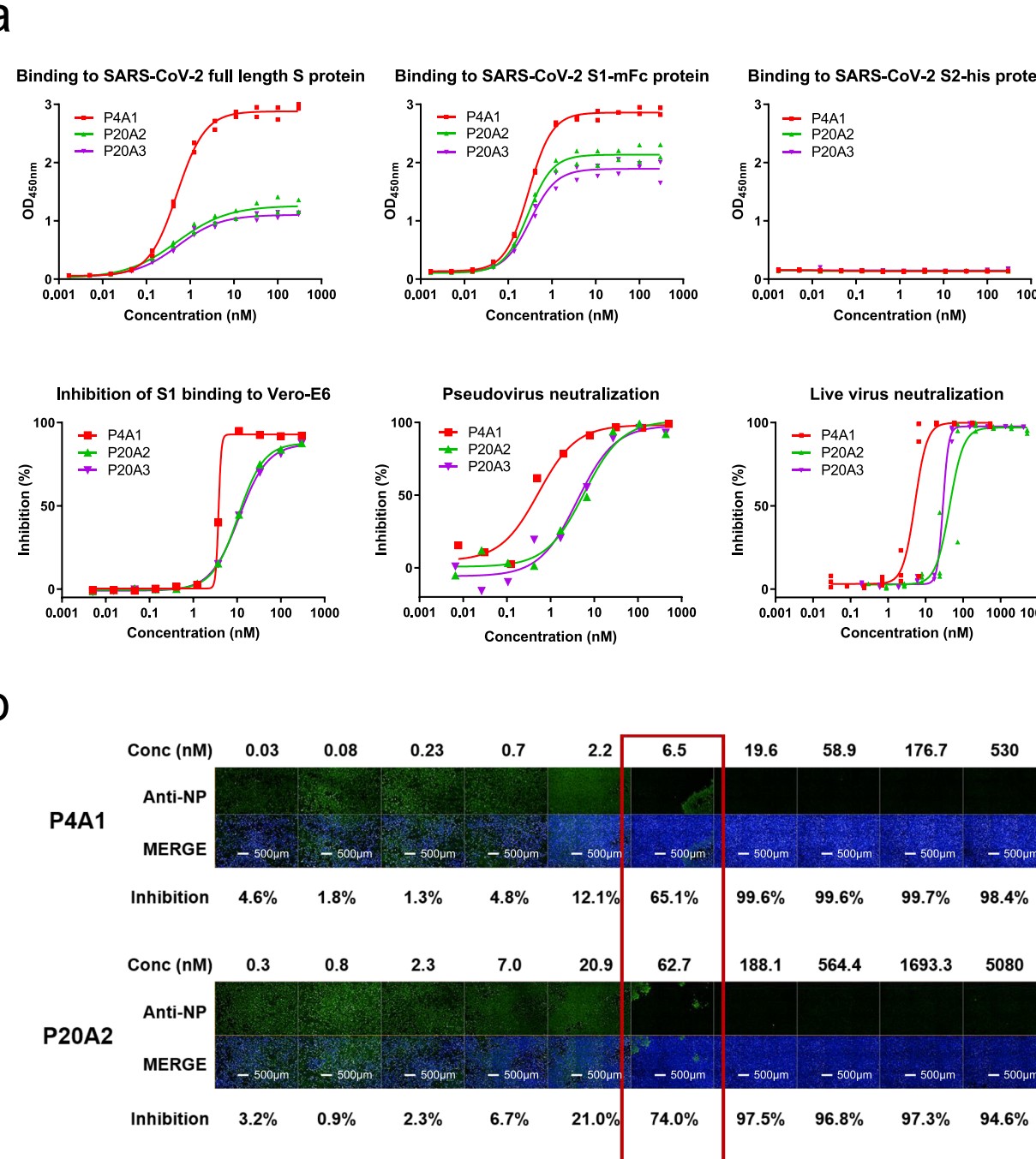

**Fig. 1 Characterization of neutralizing antibodies from convalescent patients. a** Characterization of SARS-CoV-2 S protein-specific antibodies. Upper panels: binding of antibodies to the full-length S protein, S1 protein, and S2 protein was evaluated by ELISA (in duplicates with symbols show each of the replicates). Lower left panel: blockage of the binding of SARS-CoV-2 Spike S1 protein to Vero E6 cells by antibodies evaluated by flow cytometry (data in singleton). Lower middle panel: pseudovirus neutralization assay in Huh-7 cells (data in singleton). Lower right panel: in triplicates with symbols show each of the triplicates and SARS-CoV-2 live virus neutralization assay. All experiments were repeated at least two more times (except S2 binding that was repeated one more time) with similar results. **b** Images of Vero E6 cell-infected SARS-CoV-2 treated with antibodies of different concentrations. Green (stained with SARS-CoV-2 nucleocapsid protein (NP) antibody) indicates viral infected cells and blue (Hoechst 33258) represents cell nuclei. Experiment was performed in triplicates and repeated two more times with similar results.

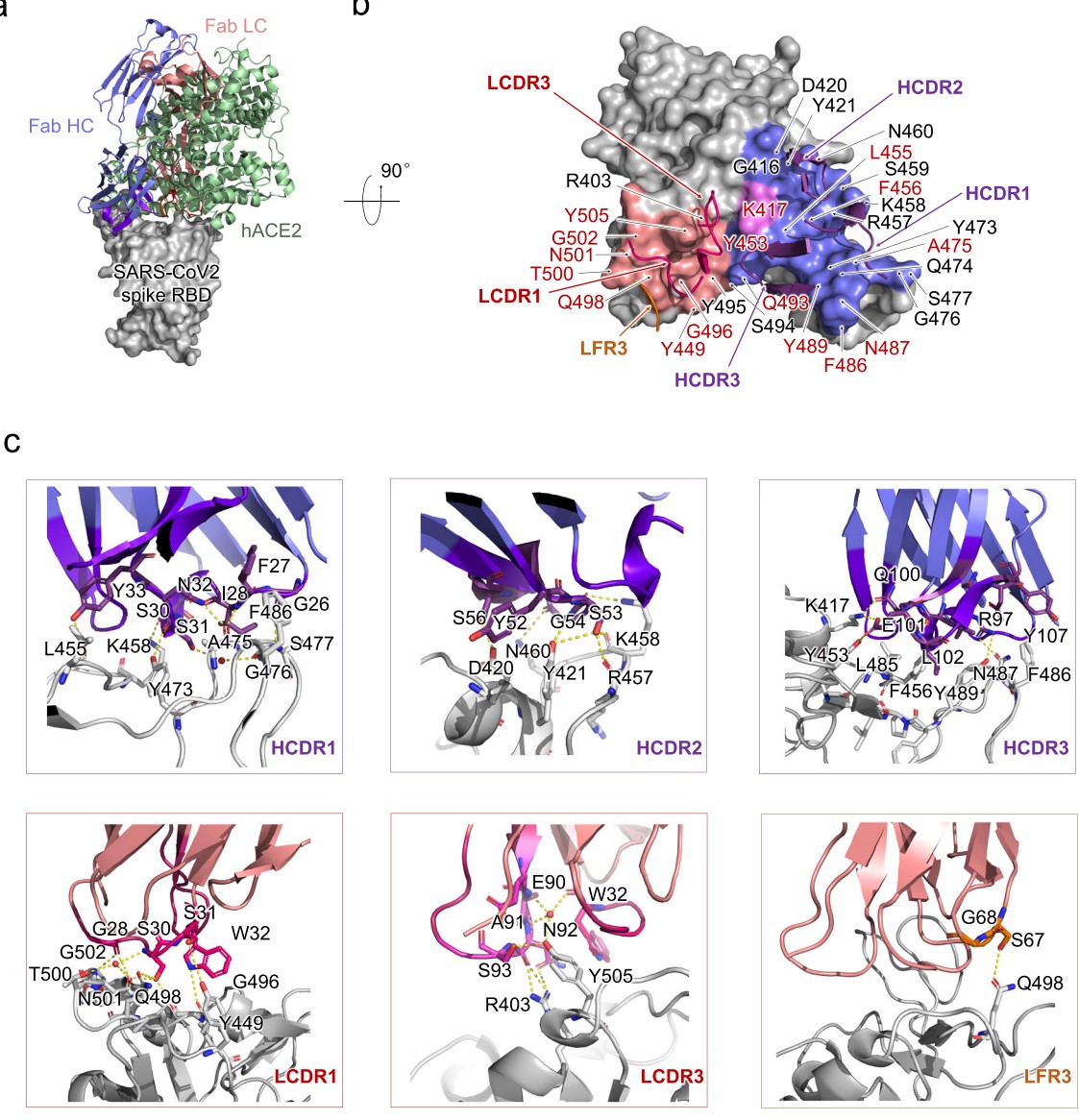

**Fig. 2 Structural analysis of P4A1 Fab and SARS-CoV-2 RBD complex. a** The overall P4A1-Fab-RBD complex structure superimposed with the hACE2-RBD complex. The P4A1 heavy chain (colored slate blue), light chain (colored salmon red), and hACE (colored pale green) are displayed in cartoon representation. The SARS-CoV-2 RBD is colored in gray and displayed in surface representation. **b** The epitope of P4A1 shown in surface representation. The CDR loops of heavy chain (HCDR) and light chain (LCDR) are colored in purple and magenta, respectively. The epitopes from the heavy chain and light chain are colored in slate blue and salmon red, respectively. The only residue K417, which contacts with both heavy chain and light chain, is colored in pink. The light-chain frame region 3 (LFR3) is colored in orange. The identical residues on RBD shared in P4A1 and hACE2 binding are labeled in red. The residues are numbered according to SARS-CoV-2 RBD. **c** The detailed interactions between SARS-CoV-2 RBD with HCDR, LCDR, and LFR3. The residues are shown in sticks with identical colors to (**b**).

P4A1 binds to the epitope on the Spike protein that locates directly at the receptor-binding motif region of SARS-CoV-2 Spike RBD, hence inducing steric hindrance to the binding of the Spike protein to receptor hACE2 (Fig. 2a). Except for residues G446 and Y449, 15 of the 17 residues identified as SARS-CoV-2 Spike RBD-hACE2 binding sites[1] are within the P4A1-targeting epitope using the distance of <4 Å as the cutoff, while Y449 is exactly 4 Å from P4A1 (Fig. 2b and Supplementary Table 2). This represents one of the most extensive coverages to date by any antibody of residues involved in the Spike-hACE2 interaction and may explain the potent neutralization activity of P4A1. As high coverage of RBD residues involved in hACE2 binding and large BSAs are also observed in other neutralizing antibodies with IGVH3–53 usage including B38[17] and CC12.1, CC12.3[16],

this may represent a class effect of the IGVH3–53 antibody family.

**Effectiveness against different Spike mutants and antibody engineering of P4A1.** With the ongoing global pandemic of COVID-19, the circulating SARS-CoV-2 mutations are subjected to strong selection to maximize the virus infection rate, some virus variant will possibly decrease its sensitivity to certain antibodies. It has been reported that single residue mutations are sufficient for the viral escape from certain neutralizing antibodies[31,32]. To explore whether P4A1 was effective against different SARS-CoV-2 Spike mutants, we tested its binding ability to 17 different RBD and 2 S1 mutants, including the N439K, Y453F, S477N, N501Y RBD variants and D614G and A222V/

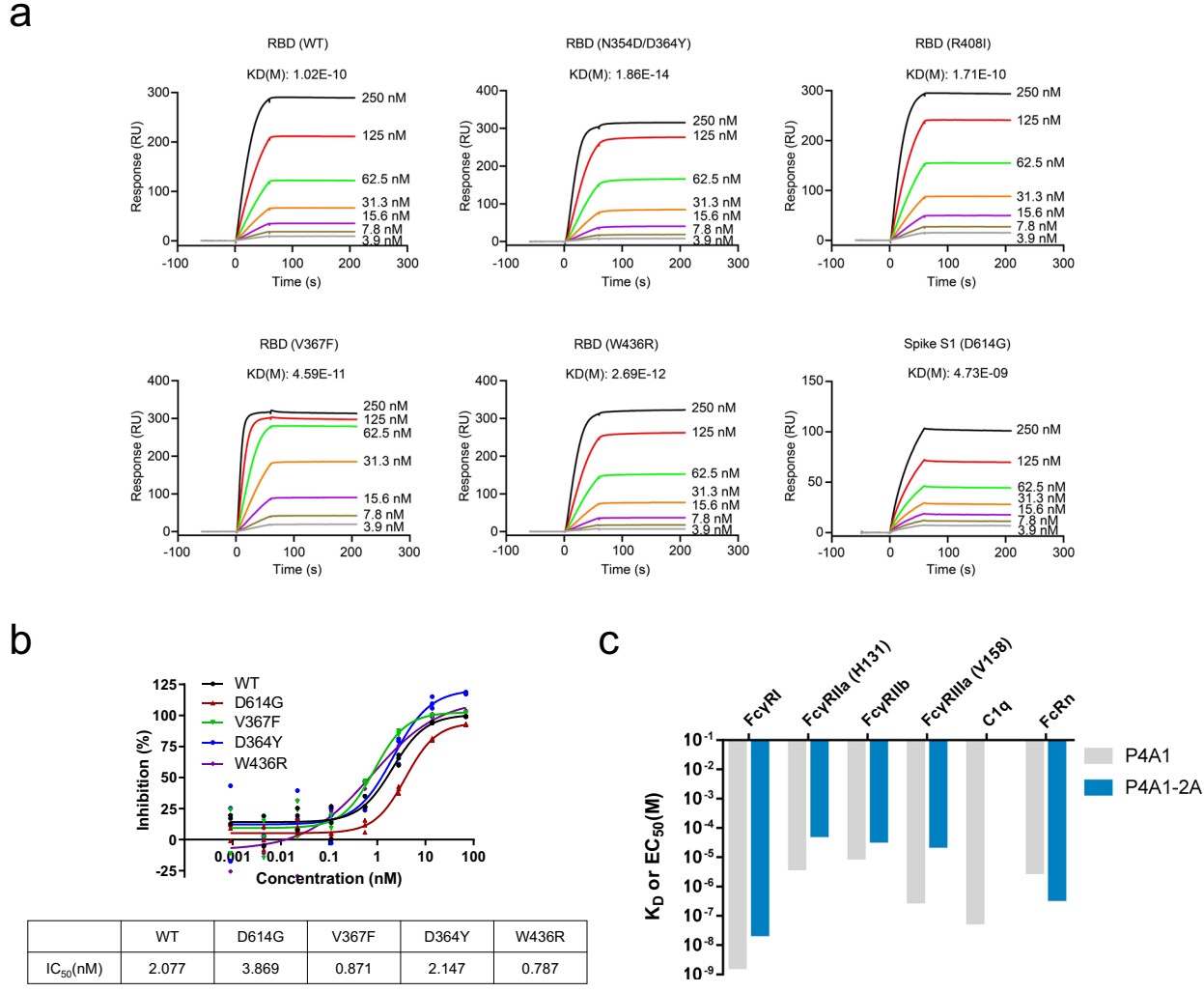

**Fig. 3 The activities of IgG4 antibody P4A1–2A to different SARS-CoV-2 S protein mutants, FcRs, and C1q. a** Binding of antibody P4A1 to SARS-CoV-2 S protein N354D/D364Y, R408I, W436R, V367F, or D614G mutants determined by surface plasmon resonance (SPR). **b** Pseudovirus neutralization assay in hACE2-overexpressing HEK293 cells. Experiment performed in triplicates with symbols represent each of the triplicates. **c** The binding affinity of P4A1 and P4A1–2A for different human FcRs and complement C1q.

D614G S1 variants that represent variants most frequently observed in the clinic. In surface plasmon resonance (SPR) (Fig. 3a and Supplementary Fig. 6a) or ELISA (Supplementary Fig. 6b) analyses, the binding of P4A1 with the variants was similar to binding with the wild-type RBD or S1 protein with similar affinities or $EC_{50}$ values except K417N variants. P4A1 also showed potent neutralization activities against pseudoviruses expressing wild-type or D614G, V367F, D364Y, or W436R variants of the Spike protein with similar $IC_{50}$s (Fig. 3b).

To maximize its clinical utility, P4A1 was further engineered as an IgG4 antibody (named as P4A1–2A) to reduce the risks of Fcγ- or complement-receptor-mediated ADE, and it was further modified to mitigate Fab-arm exchange[33], and to extend antibody half-life[34]. Consistent with its isotype and Fc engineering, P4A1–2A displayed 4–81-fold decrease in binding affinity to different FcγRs, no binding to complement C1q, and enhanced affinity for FcRn when compared to its IgG1 form P4A1 (Fig. 3c and Supplementary Fig. 7). The effect of enhanced FcRn binding is further validated by the longer serum half-life in cynomolgus monkey (Supplementary Fig. 7). The prolonged half-life is highly desired, as it will be beneficial for long-term protection in individuals at high risk of contracting SARS-CoV-2 infection during the pandemic. Furthermore, the safety of P4A1–2A was

demonstrated in a GLP toxicology study where cynomolgus monkeys were treated with 2 weekly intravenous (i.v.) infusions of up to 300 mg/kg/dose and no treatment-related adverse effects were observed in any of the tests performed, including cytokine levels, such as IL-2, IL-6, IL-10, TNF-α, and IFN-γ at different time points (Supplementary Fig. 9a). Besides, no tissue cross-reactivity was observed for 37 different normal human tissues (Supplementary Fig. 9b).

**Anti-viral efficacy in *rhesus macaques*.** The effectiveness of the engineered antibody was examined in a SARS-CoV-2 infection model in *rhesus macaques* (Fig. 4a). This and other NHP SARS-CoV-2 infection models recapitulate characteristics of COVID-19[1,35–39] and were used for evaluating the efficacy of vaccines and neutralizing antibodies for COVID-19[18,40–43]. In the study, Isotype control or P4A1–2A were administered in a single i.v. infusion 1 day after intra-tracheal virus inoculation at $1 \times 10^5$ 50% tissue-culture infectious doses ($TCID_{50}$). Consistent with previous report[1], in the isotype control (50 mg/kg) group, viral load using oropharyngeal swabs was at a high level 1 day post infection (d.p.i.), showing the colonization of virus, decreased 2 d. p.i., suggesting viral distribution, increased 3–4 d.p.i. and maintained at high level until euthanization, indicating viral

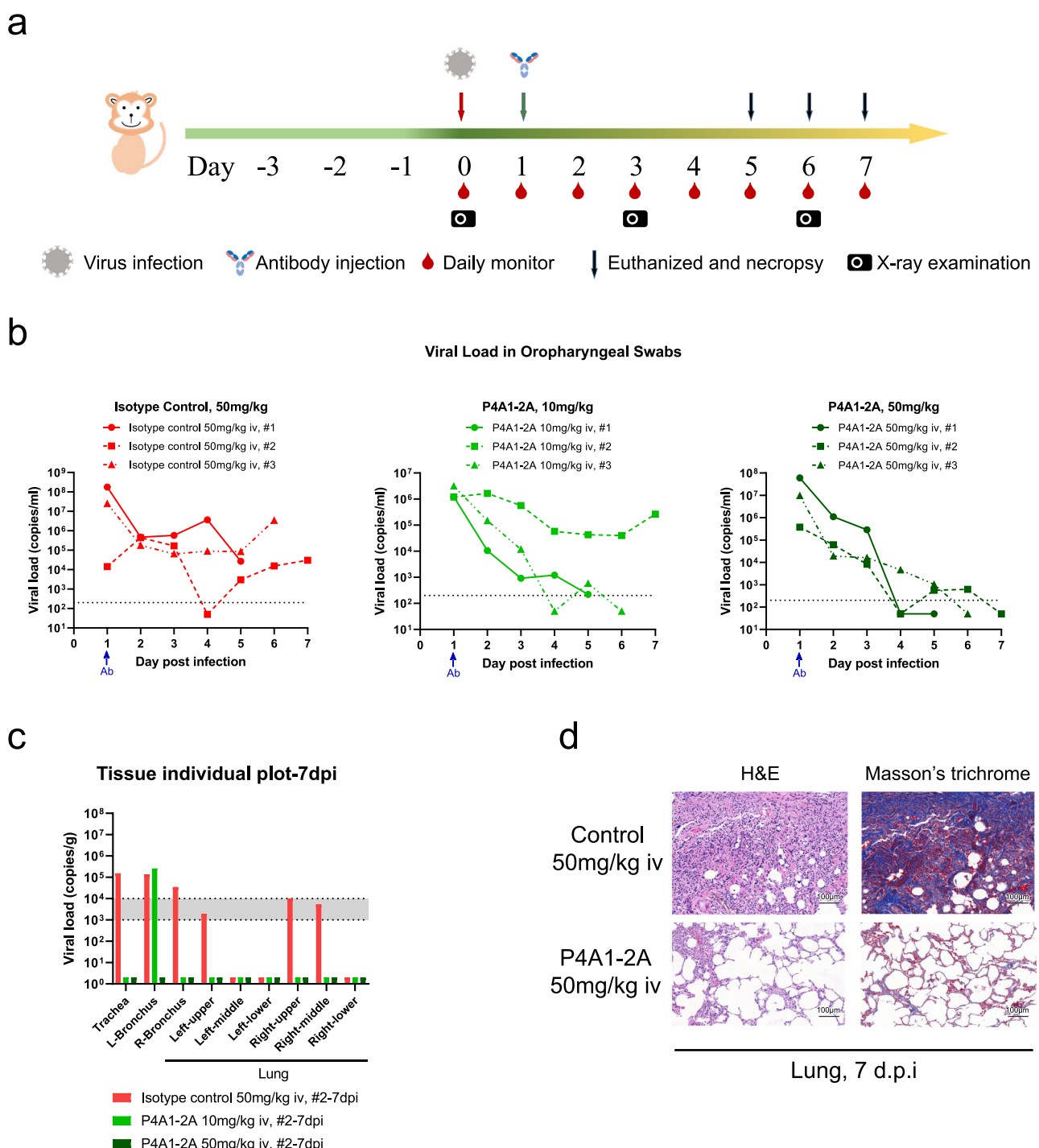

**Fig. 4 Therapeutic efficacy of in the *rhesus macaque* model of SARS-CoV-2 infection. a** Experimental design for therapeutic testing of P4A1–2A in the *rhesus macaque* (*n* = 3/group). **b** Viral load in oropharyngeal swabs tested by RT-qPCR was monitored for 7 days. **c** Viral load in the respiratory tissues (including trachea, left and right bronchus, and all six lung lobes) collected at necropsy on 7 days post infection (d.p.i., *n* = 1/group) was tested by RT-qPCR. **d** Representative images of histopathology in lung tissue from isotype control or P4A1–2A 50 mg/kg treated animals (collected at 7 d.p.i., *n* = 1/group).

replication. Importantly, in three out of three (50 mg/kg) and two out of three (10 mg/kg) P4A1-2A-treated animals, the viral load dropped continuously to a level near or below the detection limit, and viral elimination occurred on 4–5 d.p.i., indicating effective blocking of viral propagation by P4A1–2A (Fig. 4b).

At necropsy, trachea, left and right bronchus, and different lung lobes were isolated, homogenized, and tested for viral load.

In the isotype control treatment group, substantial viral load was detected in three, five, or six, respectively, out of the nine tissues examined (including consistent viral presence in trachea, left and right bronchus) in the three animals with moderate to severe lung lesions similar to a previous report[1]. In contrast, no viral load was detected 6 or 7 d.p.i. in any of the nine tissues examined and with mild to mild-moderate lung lesions in two animals treated with 50 mg/kg P4A1–2A. In the 10 mg/kg P4A1–2A-treated animals,

virus was detected in only one of the nine tissues in the animal examined 7 d.p.i. and no viral RNA was detected in any of the lung tissues in all three animals (Fig. 4c, d and Supplementary Fig. 10). Statistical analysis was not performed due to the small number of animals per group ($n = 3$), animals euthanized on different days ($n = 1$/group/day) and biological differences within the groups observed.

These results represent similar or better anti-viral efficacy in the respiratory system than the currently reported antibodies[18] with lower i.v. dose of neutralizing antibody used in similar NHP SARS-CoV-2 infection models. In addition to the efficacy of P4A1–2A, a potentially better distribution to the respiratory tract associated with the engineered IgG4 antibodies shown by previous work[34] may contribute to the total viral clearance in some of the animals. These data demonstrate that P4A1–2A is effective in vivo despite reduced FcγR effector functions.

## Discussion

High affinity neutralizing antibodies from convalescent patients may represent some of the most efficient anti-viral weapons contributing to patient recovery from viral infection. Given that not every COVID-19 patient is able to produce high-quality antibodies such as P4A1, this antibody along with antibody engineering to reduce potential safety concerns and to prolong the serum half-life may provide instant and durable protection for every individual, especially, for individuals at high risk of viral infection and/or developing severe diseases. Structural analysis demonstrates the molecular basis for the neutralizing mechanism of P4A1, whose epitope on S protein RBD includes most of the hACE2-binding residues, exhibiting potent anti-viral activities in vitro and in vivo. The broad coverage of hACE2 epitopes may contribute to the antibody's effectiveness against viral mutants, based on activities against Spike protein mutants tested, including the most prevalent N439K, Y453F, S477N, and N501Y RBD variants and D614G, A222V/D614G S1 variants, with similar binding affinities, $EC_{50}$ or $IC_{50}$s compared to wild-type protein.

Most importantly, engineered IgG4 version of P4A1 (P4A1–2A) demonstrated anti-viral efficacy in *rhesus macaque* SARS-CoV-2 infection model, despite reduced Fc effector functions. It needs to be pointed out that the study was performed with relatively small sample size ($n = 3$/group). Despite this and the biological differences observed within the group, anti-viral efficacy was clearly demonstrated based on the following: (i) none of the three animals in the isotype control group showed signs of viral clearance during the study, while five out of the six P4A1–2A-treated (10 and 50 mg/kg) animals including all three animals treated with 50 mg/kg P4A1–2A showed viral clearance; (ii) viral RNA was detected in trachea, bronchus in all three isotype-control-treated animals, and in the lung lobes of two of these animals, while two animals treated with 50 mg/kg P4A1–2A were completely viral-free in all tissues tested; (iii) clear alleviation of SARS-CoV-2 induced lung pathological changes in the 50 mg/kg P4A1–2A group compared to isotype control group; (iv) dose dependency of anti-viral efficacy by P4A1–2A treatment: 50 mg/kg group showed better viral clearance and alleviation of SARS-CoV-2 induced lung pathological changes than 10 mg/kg group. In addition to *rhesus macaque* model reported in this work, more NHP COVID-19 animal models, such as those established in African green monkey[35–37], *cynomolgus macaques*[38,39], etc., have also been developed to reflect clinical symptoms and underlying etiopathogenesis similar to those in COVID-19 patients. These disease relevant models provide alternative possibilities if we are to explore additional prophylactic application for P4A1–2A. Ultimately, the anti-viral efficacy of the antibody remains to be tested in clinical setting.

With its potent anti-viral efficacy, prolonged half-life, and its potential for a broad spectrum of viral neutralization, P4A1–2A represents not only optimized therapeutic potency, but also offers an important prophylactic option complementary to vaccine approaches to safeguard the vulnerable populations. The comprehensive characterization of this antibody provides strong evidence for its promising potential to effectively treat and prevent COVID-19 in humans.

## Method

**Ethics statements**. The study was performed in accordance with Declaration of Helsinki and all relevant ethical regulations for work with human participants. The study protocol was approved by the Ethics Committee of Xinhua Hospital affiliated to Shanghai Jiao Tong University (approval #XHEC-C-2020-006-2), Yongjia People's Hospital, and Yongjia Center for Disease Control and Prevention in Zhejiang Province. Blood samples were collected from convalescent COVID-19 patients and healthy volunteers with signed informed consent forms according to study protocol approved by IRBs.

All animal experiments were performed following Association for Assessment & Accreditation of Lab Animal Care International guidelines and all relevant ethical regulations. The study protocol was approved by the Institutional Animal Care and Use Committee of Wuhan Institute of Virology, Chinese Academy of Sciences (Ethics number: WIVA42202001), or Wuxi AppTec (Number: SZ20200529-Monkeys for PK study and SZ20200608-Monkeys for GLP Toxicology study). *Rhesus macaque* studies were conducted within the animal biosafety level 4 (ABSL- 4) facility in the National Biosafety Laboratory (Wuhan), Chinese Academy of Sciences.

**Study design and participants**. Convalescent COVID-19 patients were recruited from those who attended scheduled follow-up visits after recovering from SARS-CoV-2 infection with hospital discharge between February 11, 2020 and March 1, 2020 at Yongjia People's Hospital and Yongjia Center for Disease Control and Prevention in Zhejiang Province, China. The participants expressed willingness to participate in the study and signed Informed Consent Form. All the patients with COVID-19 were clinically diagnosed and laboratory-confirmed. A laboratory-confirmed case of COVID-19 was defined as a positive result on high-throughput sequencing or real-time reverse-transcriptase polymerase-chain-reaction assay of nasal and pharyngeal swab specimens based on the WHO interim guidance (WHO, 2020). The criteria of recovery were normal temperature for at least 3 days, obvious improvement in clinical symptoms, significant absorption of pulmonary inflammation on computer tomography scan, and negative tests for SARS-CoV-2 two times in a row with a test interval for at least 1 day. Individuals tested positive for HIV, HBV, HAV, HCV, or syphilis were excluded. All the clinical data were reviewed by a team of physicians from departments of respiratory, intensive care, and infectious diseases. Patients without the baseline assessment test were excluded. Twenty-three convalescent COVID-19 patients and one healthy volunteer were included in the study. No sample size calculation was performed. PBMCs from two donors with the highest antibody titer was used to identify potential neutralizing antibodies.

**PBMC and serum sample preparation**. Blood samples were collected using heparin as anticoagulant 3–4 days after donors were discharged from the hospital and separated into plasma and PBMCs by Ficoll-Hypaque gradient centrifugation. Plasmas and PBMCs in freezing media were stored at −80 °C. The plasma was heat-inactivated at 56 °C for 1 h before use.

**Convalescent COVID-19 patient antibody titers against SARS-CoV-2 S proteins**. The 384-well plates were coated overnight at 4 °C with PBS containing 1 μg/mL of SARS-CoV-2 Spike RBD-mFc recombinant protein (Sino Biological, Catalog #40592-V05H) or full-length S-his (Sino Biological, Catalog #40589-V08B1) or S1-mFc (Sino Biological, Catalog #40591-V05H1) or S1-his (Kactus, Catalog #COV-VM4S1). The next day the plate was washed four times with washing buffer (PBS and 0.05% Tween) and then incubated at 37 °C in blocking buffer (PBS with 2% BSA). After two washes the plate was incubated for 1 h at 37 °C with the serum or the positive control ACE2 protein (Sino Biological, Catalog #10108-H08H). The human serum samples were diluted to 1:100 in PBS + 2% BSA followed by five-fold serial dilutions. The plates were then washed four times and incubated for 1 h in blocking buffer (PBS with 0.05% Tween and 1% BSA) containing diluted (1:5000) secondary antibody (HRP-labeled mouse anti-human IgG Fc antibody, Thermo Fisher, Catalog #05-4220, clone name: HP6017) for 1 h at room temperature. Following this, the plate was washed again four times and developed in TMB substrate (Biolegend, Catalog #421101) for 5 min before stopping the reaction with the stop solution (Solarbio, Catalog #C1058).

**Convalescent COVID-19 patient B-cell profiling**. Cryopreserved PBMCs were thawed at 37 °C and centrifuged at 450 × g for 8 min. The supernatant was discarded, and the cells resuspended in 200 μL of DMEM (Gibco, Catalog #11995-

065). Following the addition of 1 μL of Dnase I (Qiagen, Catalog #79254), cells were incubated for 3 min and centrifuged again. The cell pellets were resuspended in 20 μL of FcR Blocking Reagent (Miltenyi Biotec, Catalog #130-059-901), incubated for 10 min and centrifuged. The cells were then suspended in 200 μL PBS. Then, 3 μL (1:70 dilution) of anti-CD19 (FITC labeled, eBioscience, Catalog #11-0199-42, clone name: HIB19), anti-CD27 (APC labeled, eBioscience, Catalog #17-0279-42, clone name: O323), anti-CD38 (PE labeled, eBioscience, Catalog #12-0388-42, clone name: HB7) or its isotype PE Mouse isotype control (BioLegend, Catalog #400114, clone name: MOPC-21), and FITC-labeled mouse IgG1 isotype control (eBioscience, Catalog #11-4714-41, clone name: P3.6.2.8.1) and APC-labeled mouse IgG1 isotype control (BD Biosciences, Catalog #550854, clone name: MOPC-21) were then added, and incubated for 30 min at room temperature. Following centrifugation, cells were resuspended in a 100 μL of 4% paraformaldehyde fix solution (Beyotime, Catalog #P0099–500ml). After 10 min the cells were washed twice and finally resuspended in PBS and analyzed using a Cytoflex (Beckman Coulter) flow cytometer. The median fluorescence intensity was calculated with FlowJo (version 10.6.0).

**Isolation of S protein-specific B cells**. Avi tag and His tag SARS-CoV-2 S protein was expressed in human embryonic kidney cell 293-F (Thermo Fisher, Catalog #R79007. The cell line was not authenticated.) which was tested negative for mycoplasma contamination. The S protein was purified using HISTRAP HP column (GE, Catalog #17-5248-01) and biotinylated using Biotin-Protein Ligase (GeneCopoeia, Catalog #BI001). The B cells were stained with biotinylated S protein and incubated at 4 °C for 1 h. After incubation, the cells were washed three times with PBS. Subsequently, the cells were labeled with Streptavidin microbeads (Miltenyi Biotec, Catalog #130-048-101) at 4 °C for 1 h. After the incubation, the cell suspension was loaded onto a MACS column that is placed on a magnetic field of a MACS separator. The column was washed three times so the magnetically labeled B cells were retained in the column and unlabeled cells passed through. After removing from the MACS separator, the magnetically labeled B cells were eluted. The isolated B cells were counted by using 0.4% (w/v) Trypan blue stain.

**Single-cell BCR sequencing**. Enriched S protein-specific B cells were individually co-compartmentalized in droplets with single barcoded hydrogel beads and lysis and reverse transcription (RT) reagents using a microfluidic device[15]. Droplets of ~1 nL volume were formed at 250 s-1. The droplets were collected in a 1.5 mL tube containing HFE-7500 (3M™, Catalog #Novec 7500) and 0.1% surfactant (RanBio, Catalog #008-FluoroSurfactant), UV photo-cleaved for 90 s (OmniCure ac475–365) and incubated at 50 °C for cell lysis and cDNA synthesis. RT of VH and VL mRNAs from single B cells took place in droplet using barcoded primers carrying the T7-SBS12 sequences followed by barcode and gene-specific primers sequences complementary to heavy-chain J genes and light-chain constant region sequences[44,45] (all primers used are listed in Supplementary Table 4).

The emulsion containing the barcoded cDNA was broken by adding one volume of 1H,1H,2H,2H-Perfluoro-1-octanol (Sigma-Aldrich, Catalog #370533-25G) after the droplet RT reaction finished. The pooled, barcoded cDNAs were purified with Agencourt RNA CleanUp beads (Beckman, Catalog #A63987) at a 1:1 ratio (vol/vol) twice and eluted in 60 μL DNase- and RNase-free $H_2O$. The sequencing library was generated by two-step nested PCR using GoTaq Polymerase (Promega, Catalog #M7406)[44,45]. In the first PCR, forward primers were priming on the T7 and with reverse primers priming on the VH, Vλ, and Vκ leader and framework 1 sequenced of the V genes. In the second PCR, the forward primer appends the Illumina P7 and Illumina index sequences by priming on the SBS12 sequence and the reverse primer appends the Illumina P5 and SBS3 sequence. The approximately 550 bp final PCR products were extracted by agarose gel electrophoresis (Qiagen, Catalog #28606). The constructed NGS libraries were sequenced using Illumina MiSeq PE300 that allows sequencing of the entire VH and VL domain as well as the barcode sequence (GeneScript sequencing service supplier) with data varying from 6–12 million reads per samples. The resulting FASTQ data were analyzed by a bioinformatics pipeline enable trimming, merging, barcode extraction, and clustering[45]. Briefly, paired-end reads were first trimmed at the 3' end to remove low-quality score bases then merged using the program FLASH requiring at least 10 bp overlap. The barcodes were extracted from merged reads followed by clustering requiring the DNA sharing at least 93% identify. The consensus sequence was created from clusters by aligning up to 200 sequencings using ClustalO (version 1.2), and each antibody sequence was characterized for immunoglobulin content using VDJFasta (version 2.0). We also applied a minimum number of reads 10 for VH and VL. VH–VL pairing was carried out by identifying the most abundant VH and VL consensus sequence (by the number of reads that contributed to that in each barcode cluster).

**Production of recombinant antibody**. The DNA of P4A1, P20A2, and P20A3 variable regions of the heavy and light chains were synthesized and cloned into expression plasmids containing the human IgG1 heavy-chain and kappa light-chain constant regions. The antibodies were expressed in ExpiCHO cell (Gibco, No. A29133, not authenticated) for 8 days after the co-transfection of both heavy- and light-chain expression plasmids. P4A1–2A was produced by cloning of P4A1 variable regions into expression plasmid containing human IgG4 heavy chain with

Fc modifications and expressed in CHO.K1 cells (originally from ATCC, No. CCL 61) that were qualified for use based on full characterization on identity, morphology, and freedom from adventitious agents. Antibodies were purified from cell culture supernatants using Protein-A affinity chromatography.

**Antibody binding and competition with receptor ACE2**. The binding affinity of antibodies to S protein was analyzed by ELISA. The 384-well plate (Corning, Catalog #3700) was coated overnight at 4 °C with PBS containing 30 μL 20 nM of the SARS-CoV-2 Spike S1+S2 ECD, his Tag protein (Sino Biological, Catalog #40589-V08B1), or SARS-CoV-2 Spike RBD-mFc recombinant protein (Sino Biological, Catalog #40592-V05H) or S1-mFc (Sino Biological, Catalog #40591-V05H1). The next day the plate was washed five times with washing buffer (PBS and 0.05% Tween) and then incubated 1 h at room temperature in blocking buffer (PBS with 2% BSA). After five washes the plate was incubated with a serial dilution of purified antibodies for 1 h at room temperature. The plates were then washed five times and incubated for 1 h in blocking buffer (PBS with 0.05% Tween and 1% BSA) containing Mouse anti-Human IgG Fc HRP labeled (Thermo Fisher, Catalog #05-4220, clone name: HP6017, dilution 1:5000) for 1 h at room temperature. The plate was then washed five times and incubated with TMB substrate (Biolegend, Catalog #421101) for 5 min before adding the stop solution (Solarbio, Catalog #C1058). The OD values at 450 nm wavelength were determined using Thermo MultiSkan or MD SpectraMax i3X, data were analyzed with GraphPad Prism (version 8.0.1). The experiment was repeated at least one more time with similar results or validated by other experiment.

The blocking S1 binding to cell surface ACE2 receptor was performed using flow cytometry analysis. Vero E6 cell line (ATCC, CRL-1587) was validated by CoBIOER (http://www.cobioer.com/) and tested negative for mycoplasma contamination. Then, 10 nM SARS-CoV-2 Spike S1, mFc tag protein (Sino Biological, Catalog #40591-V05H1) was incubated with a serial dilution of purified antibodies at room temperature for 1 h and then added to Vero E6 cells ($10^5$ cells per well). Rabbit anti-mouse IgG Fc-AF647 (Jackson ImmunoResearch, Catalog #315-606-046, 1:800 dilution) was then added before final wash and data acquisition with a Cytoflex flow cytometer (Beckman) and data analysis using FlowJo software (version 10.6.0). Non-linear regression was used to calculate the $IC_{50}$ of the evaluated antibodies with GraphPad Prism (version 8.0.1). The experiment was repeated two more times with similar results.

**Antibody binding to wild-type or mutant RBD/S1**. SPR experiments were performed using Biacore T200 system (GE Healthcare). In brief, experiments were performed at 25 °C in HBS-EP+ buffer. The antibody was immobilized onto a protein A sensor chip (GE Healthcare, Catalog #29139131-AA). Serially diluted SARS-CoV-2 RBD (WT, AcroBiosystems, Catalog #SPD-C52H3), RBD (V367F, AcroBiosystems, Catalog #SPD-S52H4), RBD (N354D/D364Y, AcroBiosystems, Catalog #SPD-S52H3), RBD (R408I, AcroBiosystems, Catalog #SPD-S52H8), RBD (W436R, AcroBiosystems, Catalog #SPD-S52H7), RBD (N439K, Sino Biological, Catalog #40592-V08H14), RBD (Y453F, Sino Biological, Catalog #40592-V08H80), RBD (S477N, Sino Biological, Catalog #40592-V08H46), RBD (F490S, Sino Biological, Catalog #40592-V08H41), RBD (S494P, Sino Biological, Catalog #40592-V08H18), RBD (N501Y, Sino Biological, Catalog #40592-V08H82), or SARS-CoV-2 spike S1 domain (D614G, Sino Biological, Catalog #40591-V08H3) were injected through flow cells for 60 or 180 s of association followed by a 150 or 800 s dissociation phase at a flow rate of 30 μL min$^{-1}$. Prior to the next cycle, the sensor surface was regenerated with Glycine-HCl (pH 1.5) for 30 s at a flow rate of 30 μL min$^{-1}$. KD values were calculated using the 1:1 binding kinetics model.

For ELISA, 384-well plate (Corning, Catalog #3700) was coated overnight at 4 °C with PBS containing 30 μL 20 nM of SARS-CoV-2 Spike WT or mutant RBD/S1 (mentioned above). The next day the plate was washed five times with washing buffer (PBS and 0.05% Tween) and then incubated 1 h at room temperature in blocking buffer (PBS with 2% BSA). After five washes the plate was incubated with a serial dilution of purified antibodies for 1 h at room temperature. The plates were then washed five times and incubated for 1 h in blocking buffer (PBS with 0.05% Tween and 1% BSA) containing Mouse anti-Human IgG Fc HRP labeled (Thermo Fisher, Catalog #05-4220, clone name: HP6017, dilution 1:5000) for 1 h at room temperature. The plate was then washed again five times and incubated with TMB substrate (Biolegend, Catalog #421101) for 5 min before adding the stop solution (Solarbio, Catalog #C1058). The OD values at 450 nm wavelength were determined using Thermo MultiSkan or MD SpectraMax i3X, data were analyzed with GraphPad Prism (version 8.0.1).

**Antibody neutralization activity against pseudovirus**. Huh-7 cell line (acquired from ATCC, not authenticated), which expresses ACE2 receptor, was infected with pseudovirus expressing the full length of SARS-CoV-2 Spike protein and luciferase reporter gene in the presence and absence of serial dilutions of testing antibodies. Viral entry to the cells was quantified using Britelite™ plus Reporter Gene Assay System. In separate studies, pseudoviruses encoding wild-type or different mutant S protein (GenScript, SARS-CoV-2/Wild-type (WT), SARS-CoV-2/D614G, SARS-CoV-2/V367F, SARS-CoV-2/W436R, and SARS-CoV-2/D364Y) were incubated with a serial dilution of purified antibodies (starting from 100 μg/mL, three folds dilution for eight points in triplicates) at room temperature for 1 h. The mixture

was then added to ACE2-overexpressing HEK293 cells (acquired from ATCC, not authenticated; $2 \times 10^4$ per well) cultured in DMEM containing 10% FBS in triplicate. Following infection at 37 °C 5% $CO_2$ for 48 h, luciferase activity was determined using the Promega Bio-Glo luciferase assay (Promega, Catalog #G7491) system. The dose–response curves were plotted with the relative luminescence unit against the sample concentration. Non-linear regression was used to calculate $IC_{50}$ using GraphPad Prism 6. WT pseudovirus was tested two more times with similar results.

**Live virus assay**. Microneutralization assays toward live virus were performed as previously described[46] with slight modifications. The Vero E6 cell line (ATCC; CRL-1586, Lot #60526234; this clone was not authenticated) was tested negative for mycoplasma contamination using a commercial EZ-PCR$^{TM}$ Mycoplasma Test kit (Biological Industries, Beit-Haemek, Israel; 20-700-20; Lot #1251719). Briefly, SARS-CoV-2 (strain BetaCoV/Wuhan/WIV04/2019 preserved in National Virus Resource Center under the accession number: IVCAS 6.7512) was mixed with equivalent volume of culture medium containing serially diluted antibodies and incubated at 37 °C for 1 h. Then, Vero E6 cells seeded in 96-well plates were incubated with a pre-incubated mixture contained virus at 100 $TCID_{50}$ per well and diluted antibodies and were further sustained at 37 °C for 48 h. Subsequently, cells were fixed with 4% paraformaldehyde diluted in PBS for 15 min and penetrated by 0.25% Triton-X 100. After three washes with PBS, cells were blocked at 37 °C for 1 h using PBS containing 5% BSA, then incubated with in-house prepared anti-SARS-CoV-2 nucleocapsid protein (NP) rabbit serum produced internally at 1:1000 dilution as primary antibody and Goat Anti-Rabbit IgG H&L (Alexa Fluor® 488) (1:500 dilution, Abcam, Cambridge, UK; ab150077; Lot #GR3244688-2) as the secondary antibody. Cell nuclei were stained using Hoechst 33258 (Beyotime, Shanghai, China; C1018). Images were taken, and numbers of nuclei and cells infected with viruses were counted, respectively, using an Operetta CLSTM system (PerkinElmer, Waltham, USA). Inhibition was calculated by (total nuclei-infected cells)/total nuclei × 100%. $ND_{50}$ were calculated with GraphPad Prism 8.0. The experiment was repeated two more times with similar results.

**RBD protein expression and purification**. The codon optimized wild-type cDNA of SARS-CoV-2 RBD (residues 333–530) was synthesized. The SARS-CoV-2 RBD with a C-terminal 8×His tag for purification was cloned into pAcgp67 vector with BamH I and Not I restriction sites using the cloning primers. The sequences of the primers are listed in Supplementary Table 4. The accuracy of the inserts was verified by sequencing.

The SARS-CoV-2 RBD was expressed using the Bac-to-Bac baculovirus system. The construct was transformed into bacterial DH5α component cells, and the extracted bacmid was then transfected into Sf9 cells using Cellfectin II Reagent (Invitrogen). The low-titer viruses were harvested and then amplified to generate high-titer virus stock. The viruses and Endo H, Kifunensine were co-infected Hi5 cells at a density of $2 \times 10^6$ cells/mL. The supernatant of cell culture containing the secreted removal of glycosylated RBD was harvested 72 h after infection, concentrated and RBD was captured by Ni-NTA resin (GE Healthcare). The resin was washed five to six times with 30 mL of wash buffer (25 mM Tris, 150 mM NaCl, 40 mM imidazole, pH 7.5), the target protein was eluted with elution buffer containing 25 mM Tris, 150 mM NaCl, 500 mM imidazole, pH 7.5. The protein was further purified on a Superdex S75 (GE Healthcare) column equilibrated with 25 mM Tris, 150 mM NaCl, pH 7.5. SDS-PAGE analysis revealed over 95% purity of the final purified recombinant protein. Fractions from the single major peak were pooled and concentrated to 15 mg/mL RBD were collected.

**Crystallization**. The SARS-CoV-2 RBD protein and P4A1-Fab fragment were mixed at a molar ratio of 1.5:1. The mixture was incubated on ice for 1 h and further purified by Superdex S75 (GE Healthcare). Then, 7 and 10 mg/mL of SARS-CoV-2 RBD/Fab proteins were used for crystal screening by vapor-diffusion sitting-drop method at 16 °C, including the Index, Crystal Screen, PEG/Ion, SaltRX from Hampton Research, and wizard I–IV from Emerald BioSystems.

The rode-like crystals appeared after 2 days at the mother liquid containing 20% w/v PEG 3350, 0.2 M potassium citrate tribasic. Further optimization with additive and hanging-drop vapor-diffusion method was performed, the final optimized diffraction crystals at the mother liquid containing 20% w/v PEG 3350, 0.2 M potassium citrate tribasic by the hanging-drop vapor-diffusion method. Crystals were dehydrated and cryo-protected in 4M Sodium formate solution and cooled in a dry nitrogen stream at 100 K for X-ray data collection.

**X-ray data collection, processing, and structure determination**. Diffraction data were collected at the Shanghai Synchrotron Radiation Facility BL17U1 (wavelength, 0.97915 Å) at 100 K. All data sets were processed using the HKL3000 package[47]. Structures were solved by molecular replacement using PHASER[48] with the SARS-CoV-2 RBD structure (PDB ID: 6M0J)[43] and the structures of the Fab fragment available in the PDB with the highest sequence identities. The initial model was built into the modified experimental electron density using COOT (Version 0.9.4)[49] and further refined in PHENIX (Version 1.19)[50]. Model geometry was verified using the program MolProbity. Structural figures were drawn using the program PyMOL (Version 1.8) (http://www.pymol.org). Epitope and paratope

residues, as well as their interactions, were identified by accessing PISA (http://www.ebi.ac.uk/pdbe/prot_int/pistart.html) at the European Bioinformatics Institute.

**Antibody binding affinity to FcγRs and C1q**. The binding affinity of P4A1–2A as well as its IgG1 form for different human FcγRs, FcRn was tested by SPR using Biacore 8K (GE Healthcare). FcγRI, FcγRIIa (H131), FcγRIIb, and FcγRIIIa (V158) (all from AcrobioSystems) were captured on the activated CM5 sensor chips, followed by flow-through of ten concentrations of serial diluted P4A1 (IgG1) or P4A1–2A (engineered IgG4) antibodies. For FcRn binding, P4A1 or P4A1–2A were captured on the activated CM5 sensor chips followed by flow-through of ten concentrations of serial diluted FcRn (AcrobioSystems). The sensor chips were regenerated by flow-through of 10 mM glycine (pH 1.5). Experiments were not repeated as controls were consistent with historical data and literature, and/or results of testing articles are supported by other experiments.

Antibody binding to C1q was tested by ELISA, with P4A1 or P4A1–2A antibodies to coat the plate overnight at 4 °C, followed by incubation with 11 concentration of half-log titrated human C1q (Complement Technology), then incubation with secondary antibody Sheep anti-human C1q Ab-HRP (Complement Technology, Catalog #CPBT-65026SH; dilution 1:300), before TMB substrate was added and absorbance at 450 nm was determined using a SpectraMax Plus384 microplate reader. Experiments were not repeated as controls were consistent with historical data and/or literature.

**Pharmacokinetic (PK) analysis**. Six (three/sex) naïve cynomolgus monkeys were enrolled in the study and administered with P4A1–2A at 10 mg/kg by a single i.v. infusion. Blood sample PK analyses were performed at pre-dose, and different time points after the treatment. Serum concentrations of P4A1–2A were determined using a validated ELISA method by an independent laboratory.

**Safety evaluation**. Cynomolgus monkeys (3–5 years old, 2.2–3.6 kg for females and 2.3–5.2 kg for males, sourced from GuangDong Blooming-Spring Biological Technology Development Co., Ltd.) were randomly assigned to three groups of five/sex/group and treated with P4A1–2A at 0, 50, or 300 mg/kg/dose on Day 1 and Day 8 by i.v. infusion. Animals were monitored daily with tests performed, blood samples collected at different time-point for various testing. Animals were euthanized on Day 15 (dosing phase) or Day 71 (recovery phase) and examined thoroughly for any potential toxicity. Serum concentration of P4A1–2A and cytokines were determined using validated ELISA methods by independent laboratories.

**In vivo efficacy in SARS-CoV-2 infection model in rhesus monkey**. Nine rhesus monkeys (RMs, three males, six females; 6–7 years of age, 5.3–7.3 kg, sourced from Hubei Tianqin Biotech Company) were inoculated with SARS-CoV-2 virus (BetaCoV/Wuhan/WIV04/2019)[46] at $1 \times 10^5$ $TCID_{50}$ intratracheally under anesthesia on Day 0. Animals were randomized into three groups (one male and two females in each group) and received a single i.v. treatment of Isotype control W332-1.80.12.xAb.hIgG4 (sourced from Wuxi Biologics, Co. Ltd.) at 50 mg/kg, or P4A1–2A at 10 or 50 mg/kg, respectively, 1 day after viral infection. The RMs were observed twice daily with detailed recording of clinical signs. Swab samples of the oropharyngeal, nasal turbinate, and rectal regions, as well as whole blood (in $K_2$EDTA tubes), were collected at 0–7 d.p.i. To confirm the pathogenesis and injury in the respiratory system, one animal from each treatment groups was sacrificed at 5–7 d.p.i., respectively. The trachea, right bronchus, left bronchus, all six lung lobes, and other tissue organs were collected on the day of euthanization for various pathological, virological analyses. Viral loads in swabs and tissues were determined by RT-qPCR and pathological examination. Briefly, total RNA was extracted from organs with an RNeasy Mini Kit (Qiagen, USA) and PrimerScript RT Reagent Kit (TaKaRa, Japan). The forward and reverse primers targeting the SARS-CoV-2 NP gene for RT-PCR (Supplementary Table 4) were 5'-GGGGAACTTCTCCTGCTAGAAT-3' and 5'-CAGACATTTT GCTCTCAAGCTG-3', respectively. RT-PCR was performed under the following reaction conditions: 42 °C for 5 min, 95 °C for 10 s, and 40 cycles of 95 °C for 10 s and 60 °C for 30 s. Viral load determination was repeated once with similar results. Histopathology evaluation was performed independently by investigator and a pathologist with similar findings.

Animal studies were not repeated as controls were consistent with historical data and/or literature.

**Validation of the antibodies**. All antibodies were validated and dilution optimized using positive cells (antigen-transfected 293T cells (acquired from Chinese Academy of Sciences; Catalog #GNHu17; authenticated by STR analysis) or PBMCs) before performance of the studies.

**Reporting summary**. Further information on research design is available in the Nature Research Reporting Summary linked to this article.

## Data availability
The atomic models generated from X-ray crystallographic studies of the P4A1-RBD complex has been deposited at the Protein Data Bank (PDB, http://www.rcsb.org/) under accession codes PDB 7CJF. The authors declare that all data are available within the article and its Supplementary Information files, or are available from the authors upon request. All reagents and information presented in this study are available from corresponding authors upon reasonable request. Source data are provided with this paper.

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

## Acknowledgements
We thank the staffs at National Biosafety Laboratory, Center for Instrumental Analysis and Metrology (both at Institute of Virology, Chinese Academy of Sciences, Wuhan, Hubei Province) and BL-17U1, Shanghai Synchrotron Radiation Facility (Shanghai). We thank Cheryl Cui and Jeff He for their critical role in setting up collaboration among different institutions, and Ning Zhang from Red Avenue Foundation for her support. This work was supported by the National Program on Key Research Project of China (2018YFE0200400, 2018YFA0507203, 2017YFC840300, 2017YFA0504801, and 2020YFC0845801), the National Natural Science Foundation of China (NSFC)

(31670731, 31870733), Projects of International Cooperation and Exchanges NSFC (grant no. 81520108019), Medical and Industrial Cross Research Foundation of Shanghai Jiao Tong University (YG2020YQ27), Science and Technology Innovation Achievements and Team Building Foundation of Nankai University (grant no. ZB19500403, ZB19100123, and 63201101), and Emergency Key Program of Guangzhou Laboratory (Grant No. EKPGL2021008).

## Author contributions

Z. R., K. S., L. S., H.-K. Z., Z.-Y. L., and Y. G. conceived the project. L.-S. H., X.-X. Z., W.-J. Z., H.-Y. H., and K. S. collected the convalescent PBMC and serum and are responsible for human-material-related work. Y. G., L.-S. H., G.-S. Z., B.-Q. S., M.-J. C., J. W., M.-F. F., Y.-R. L., B. L., D. F., W. Z., Y.-H. W., Y. C., J.-W. L., and C. Z. performed gene construction, protein expression and purification, and crystal screening and optimization. Y. G., X. L., Q.-S. W., Q. Z., F. Y., A. S., and A. G. G. performed the collection of X-ray diffraction data and structure determination. L.-S. H., G.-S. Z., B.-Q. S., S.-S. W., Y.-Y. L., M.-M. L., C.-J. P., D. C., Y. W., and X.-D. Z. performed the in vitro binding assays and analysis data. S. S., Y.-H. F., and F. D. performed live virus neutralization assays. C. S., Y. F. Y., X. H., G. G., Y. P., J. M., and Z.-M. Y. performed animal assays. Y. G., H.-K. Z., L. S., H. Z., Z. R., K. S., and F. A. designed the experiments and wrote the manuscript. All authors read and approved the contents of the manuscript.

## Competing interests

L. S., B.-Q. S., H. Z., M.-J. C., Y.-Y. L., L. S. H., and K. S. are the inventors of the pending patent application filed on the reported antibodies. L. S., B.-Q. S., H. Z., M.-J. C., J. W., M.-F. F., C.-J. P., M.-M. L., Y.-Y. L., S.-S. W., Q. Z., and F. A. are employees of HiFiBio (Hong Kong) Ltd. Y.-R. L. is an employee of Wuxi Biologics Ltd. Other authors declare no competing interests.
