## [Peer Review File · Nature Communications]

REVIEWER COMMENTS

Reviewer #1 (Remarks to the Author):

In this work, the authors identify a SARS-CoV-2 neutralizing Ab from convalescent patient serum, which they evaluate using X-ray crystallography, in vitro functional measurements (binding, as well as pseudovirus and live virus neutralization), and a rhesus macaque in vivo efficacy study. They engineered their mAb as an IgG4 antibody, which is an interesting approach if the goal is to reduce Fc receptor binding and effector function. The mAb showed remarkable neutralizing activity, but not particularly high compared to other published mAbs, especially if compared to several currently undergoing clinical trials.

The study is well executed, but I have one major request for additional experiments prior to publication:

The authors infer from their structural data that P4A1 has “exceptional spike binding coverage” based on the proportion of the RBM that overlaps with the epitope. This claim is based on the assertion that mutations in the RBM are rare in clinical isolates (line 136). However, this is not true, and in fact the two most commonly observed RBD variants to date (S477N and N439K) are RBM mutations. The authors should take a closer look at the sequence databases, pick out the top 5-10 most frequent RBM variants plus a few other high frequency RBD variants, and test these for binding and/or pseudovirus neutralization before they are able to make a conclusion about breadth. In this study Authors have already tested a few RBD variants, but none of these variants contain mutations near the epitope so it is not surprising that they observed no loss of potency.

Based on the Introduction, the authors appear to be evaluating SARS-CoV-2 circulating sequences based on an analysis pipeline built at LANL (<https://cov.lanl.gov/content/index>). However, the correct reference for clinical isolate sequences is the underlying database from GISAID (gisaid.org). It appears that the reliance on the LANL website has prevented the authors from realizing the extent of the mutations occurring in the RBD and RBM, as the “Tracking Mutations” tool on the LANL website only includes 19 mutations in the spike protein, when in fact there are dozens of additional positions where mutations have been observed.

Overall, this study presents very nice data but requires additional experiments to support the claim of breadth, and the novelty is not high.

Reviewer #2 (Remarks to the Author):

The manuscript entitled “A SARS-CoV-2 neutralizing antibody with exceptional spike binding coverage and optimized therapeutic potentials” by Guo et al. describes the identification, characterization, and optimization of an anti-SARS-CoV-2 human monoclonal antibody, P4A1. The authors then assessed the efficacy of P4A1 in a rhesus macaque clearance model of SARS-CoV-2. The manuscript is relevant and timely given the recent use and apparent success of human anti-SARS-CoV-2 monoclonal antibodies from Regeneron and Eli Lilly in treating COVID-19 cases under certain conditions.

Major Comments

1. The main concern with the paper is the appropriateness of the nonhuman primate model used and small number of animals employed for the efficacy study. The rhesus macaque model is essentially a SARS-CoV-2 clearance model where animals do not appear to show much in the way of clinical signs of COVID-19. What was the rationale for using this model versus the African green monkey model

which does appear to recapitulate human COVID-19 (e.g., Woolsey et al., Nat Immunol. 2020 Nov 24. doi: 10.1038/s41590-020-00835-8; Blair et al., bioRxiv 2020.06.18.157933; Cross et al., Virol J. 2020 Aug 18;17(1):125; Blair et al. Am J Pathol. 2020 Nov 7:S0002-9440(20)30497-1)? Also, the group sizes were only n = 3 per group and because one animal per group was euthanized on days 5, 6, and 7, respectively it is only n = 1 per group when comparing parameters like tissue viral load or pathology. And the only time where viral load could be compared in swabs at n = 3 was the time points at or before day 5. There is clearly no way to achieve any statistical significance in results among the groups.

2. Why did the authors not assess BALS fluid which is the best indicator of viral replication in the upper respiratory tract of nonhuman primates and is commonly assessed in most studies?

Minor Comment

1. Why were the safety and PK tests done in cynomolgus monkeys and the efficacy tests done in rhesus monkeys?

Reviewer #3 (Remarks to the Author):

This paper describes yet another neutralizing antibody against SARS-CoV-2 that could be used as therapeutic to treat COVID19 infections.

The authors used a complementarity of techniques to isolate and characterize (using SPR and X-ray crystallography) the most potent antibody they identified. They then changed its isotype to reduce potential ADE and tested its therapeutic effect in cynomolgous monkeys.

The paper needs to be edited for better readability (english grammar) - see some suggestions below - but they are really all over the places.

Some references appear to be missing - with the vast number of publications on this subject, the authors could do a better job including all the work (similar to their) that has already been published.

The method used to identify the antibodies is not really "high-throughput single-cell B-cell receptor sequencing approach" since it involves using a "probe" to isolate the antibodies first and then sequence while what they mention could refer to just sequencing the B cell receptor. Maybe the authors can clarify this.

The statement (line 134) "This represents one of the most extensive coverages of residues involved in the Spike hACE2 interaction to date and may explain potent neutralization activity for P4A1." can be better explained in the context of the other similar antibodies (VH3-53) that have been isolated and bind the same site (from fig S4). It seems unclear if all the antibodies of that "class" have the same potency.

As such the title of the paper is misleading - it does not appear that this particular antibody has more exceptional binding coverage compared to the other published Mabs that bind the same site?

Regarding the structural analysis, the authors could add a PISA style table to include the interface. Also it seems that the geometry of the NAG in the structure is wrong (from the PDB validation).

It would have been interesting to compare PA41 IgG1 to the engineered PA41-IgG4 in monkeys as well as test the prophylactic role of the antibody.

Thus again the title seems misleading as it is unclear if indeed the antibody therapeutic potential was optimized since there is no comparison to the IgG1 (the non optimized?).

Examples where english writing could be improved and references added

line 75 - add references

line 83 change “also” to “was”

line 84: and raised a concern - rewrite that sentence..

line 86 - mutations rather than mutation sites.

something seems to be missing in the last sentence of the introduction

line 96 - reads weird

SARS-CoV-2 S protein binding B cells from PBMCs were enriched 97 with biotinylated S protein conjugated magnetic beads as probes as described

line 102 ND50 values of 5.212, 43.79, and 29.03 nM, shouldn't this be in nM/mL?

line 105 - could add more references

line 128 - P4A1 epitope

line 135- change broad coverage - here the authors likely mean the large overlap between ACE2 and P401 epitope on the RBD...?

line 137 - is there a reference?

line 139 - missing “and thus are expected to..”

line 140 rewrite (it was supported by binding experiments...maybe add the mutations

line 176 - word seems to be missing

Fig S4 - superposition...

Dear Reviewers,

The authors greatly appreciate the constructive critiques from the reviewers. The manuscript has been revised according to the reviewers' comments and we believe this has resulted in improved quality.

As suggested, we performed binding assays against 10 mutants to support the claim on the breadth of P4A1, including the most prevalent RBD mutants N439K, Y453, S477N and N501Y, the result has been added to FigureS4B and FigureS5; PISA style table was provided as Table S3. In addition, we updated figure3C to better illustrate the engineering effect of P4A1-2A.

In this round of review, reviewers request more animal model to support the conclusion. As you may know, the protection test of NAb in non-human primate model need at least three months. We worry that this may significantly slow down the appearance of this work to readers in such a fast-moving field. Moreover, the protection test procedure in this work is widely accepted in this field, as reported by Shan et al, Cell Res. 2020. 30:670-677, van Doremalen et al. Nature. 2020. 586:578-582; Feng et al. Nat Commun. 2020. 11:4207; Zost et al. Nature. 2020. 584: 443-449; Shi, R. et al. Nature. 2020. 584:120-124; Wang et al. Nat Commun. 2020. 11:5752). Therefore, we believe this result could fully support our conclusion and would not like to provide more animal model in this revision with your and reviewers' permission.

The authors feel that the reviewers' comments have been adequately addressed and would like to submit the revised manuscript entitled 'A SARS-CoV-2 neutralizing antibody with extensive spike binding coverage modified for optimal therapeutic outcomes' to be considered for publication in the journal of Nature Communications.

Here are our point-to-point response to reviewers' comments (in blue)

Reviewer #1:

The authors infer from their structural data that P4A1 has "exceptional spike binding coverage" based on the proportion of the RBM that overlaps with the epitope. This claim is based on the assertion that mutations in the RBM are rare in clinical isolates (line 136). However, this is not true, and in fact the two most commonly observed RBD variants to date (S477N and N439K) are RBM mutations. The authors should take a closer look at the sequence databases, pick out the top 5-10 most frequent RBM variants plus a few other high frequency RBD variants, and test these for binding and/or pseudovirus neutralization before they are able to make a conclusion about breadth. In this study Authors have already tested a few RBD variants, but none of these variants contain mutations near the epitope so it is not surprising that they observed no loss of potency.

Thanks for this suggestion, the authors agree with the reviewer that additional mutant testing is important to support the claim on the breadth of the antibody, we performed binding assays and revise some sentences accordingly.

Due to the limited availability of reagents and low abundance associated with some of these mutants before the manuscript preparation, mutants other than the ones included in the original submission were not tested at the time. Since the initial drafting of the manuscript, we were carefully monitoring RBD and other S protein mutations included in publications and online databases including GISAID. As of December 13 2020, N439K, Y453F, S477N and N501Y are the most prevalent mutations in the RBD, with global frequencies of 1.5, 0.55, 6 and 0.58% respectively (https://cov.lanl.gov/components/sequence/COV/int_sites_tbls.comp, analyzed from the GISAID sequences collected from Dec 2019 to Dec 13 2020).

Since the submission of the manuscript, S477N, N439K, N501Y, Y453F, S494P and F490S RBD mutants became commercially available. We have acquired these mutants and performed binding assays (pseudoviruses encoding these spike variants are not yet available). P4A1 displayed similar binding to these RBD mutants as to WT RBD with similar KD or EC₅₀. The results are described and included in Figures S5 in the revised manuscript.

Additional mutant proteins are being purified or custom-made, and will be used in binding experiments as soon as they are available. However, as this will be an on-going effort with our continued monitoring of SARS-CoV-2 viral landscape, and that our efforts so far included N439K, S477N, Y453F, and N501Y as the most prevalent RBD mutants now and sitting on the edge of the P4A1 epitope, we would really appreciate the manuscript to be considered for publication by Nature Communications ahead of future data generation because of the time-sensitive nature of the subject and potential benefit to the scientific community.

Based on the Introduction, the authors appear to be evaluating SARS-CoV-2 circulating sequences based on an analysis pipeline built at LANL (<https://cov.lanl.gov/content/index>). However, the correct reference for clinical isolate sequences is the underlying database from GISAID (gisaid.org). It appears that the reliance on the LANL website has prevented the authors from realizing the extent of the mutations occurring in the RBD and RBM, as the “Tracking Mutations” tool on the LANL website only includes 19 mutations in the spike protein, when in fact there are dozens of additional positions where mutations have been observed.

The authors thank the reviewer for this important information.

The authors chose the LANL pipeline to extract the SARS-CoV-2 mutations from the GISAID database following the publication reporting the prevalence and higher infectivity of the D614G mutation (DOI:10.1016/j.cell.2020.06.043). The LANL

pipeline performs daily extraction from the GISAID database. We have downloaded the mutation data tables using the “Tables of mutating sites” tool, that includes all mutations rather than the “Tracking mutations” tool. By analyzing all reported sequences from GISAID from Dec 24 2019 to Dec 13 2020 with the LANL pipeline (described in [DOI:10.1016/j.cell.2020.06.043](https://doi.org/10.1016/j.cell.2020.06.043)), mutations (with frequency ≥ 0.3 % for non-ACE2 interface residues and $\geq 0.1\%$ for ACE2 interface residues) to about 172 residues in the Spike protein have been identified. The table below indicates the RBD mutations with the highest abundance found as of December 13 2020:

ori	pos	mut	global count	global frequency
N	439	K	3256	0.0155017
N	439	X	120	0.0005713
Y	453	F	1158	0.0055132
Y	453	X	10	0.0000476
S	477	N	12785	0.0608688
S	477	I	131	0.0006237
S	477	X	66	0.0003142
S	477	R	21	0.0001
S	477	G	2	0.0000095
S	477	T	2	0.0000095
N	501	Y	1217	0.0057941
N	501	T	30	0.0001428
N	501	X	17	0.0000809
N	501	S	4	0.000019
N	501	R	1	0.0000048

As suggested, we updated our description in the introduction section accordingly.

Reviewer #2:

Major Comments

1. The main concern with the paper is the appropriateness of the nonhuman primate model used and small number of animals employed for the efficacy study. The rhesus macaque model is essentially a SARS-CoV-2 clearance model where animals do not appear to show much in the way of clinical signs of COVID-19. What was the rationale for using this model versus the African green monkey model which does appear to recapitulate human COVID-19 (e.g., Woolsey et al., Nat Immunol. 2020 Nov 24. doi: 10.1038/s41590-020-00835-8; Blair et al., bioRxiv 2020.06.18.157933; Cross et al., Virol J. 2020 Aug 18;17(1):125; Blair et al. Am J Pathol. 2020 Nov 7:S0002-9440(20)30497-1)? Also, the group sizes were only $n = 3$ per group and because one animal per group was euthanized on days 5, 6, and 7, respectively it is only $n = 1$ per group when comparing parameters like tissue viral load or pathology. And the only time where viral load could be compared in swabs at $n = 3$ was the time points at or before day 5. There is clearly no way to achieve any statistical significance in results among the groups.

Thank you for this suggestion, the authors appreciate the African green monkey SARS-CoV-2 infection model as a relevant COVID-19 animal model. The rhesus macaque SARS-CoV-2 infection model is also recognized as a relevant model displaying COVID-19 features that include viral replication and spread, as well as radiological and pathological changes in the lungs (Shan et al, *Cell Res.* 2020. 30:670-677). The rhesus model has been used for evaluating the efficacy of vaccines and neutralizing antibodies for COVID-19 (van Doremalen et al. *Nature.* 2020. 586:578-582; Feng et al. *Nat Commun.* 2020. 11:4207; Zost et al. *Nature.* 2020. 584: 443–449; Shi, R. et al. *Nature.* 2020. 584:120-124; Wang et al. *Nat Commun.* 2020. 11:5752). In isotype control group in the study included in the manuscript, SARS-CoV-2 viral load was maintained at high level, with viral RNA isolated from multiple respiratory organs in all 3 animals, and macroscopic (pictures below) and moderate to severe microscopic (Figure 4 and Figure S8) pathological changes (including hemorrhage, edema, extensive inflammatory cell infiltration, fibrosis) were observed in the lung tissues.

The sample size was relatively small (n=3) to adhere to the AAALAC 3R principle ('Reduce' in this case), as well as the extensive procedures (especially during necropsies with many samples/organs to be harvested according to SOPs of the BL4 facility) and limited availability of rhesus macaque prevented a much larger study in this model. Despite this small sample size, we believe that the anti-viral efficacy was clearly demonstrated based on the following: **i)** none of the 3 animals in the isotype control group showed signs of viral clearance during the study, while 5 out of the 6 P4A1-2A-treated (10 and 50 mg/kg) animals and all 3 animals treated with 50mg/kg P4A1-2A showed viral clearance. This difference between P4A1-2A-treated (10 and 50 mg/kg or 50 mg/kg) group compared to isotype control reached statistical significance by contingency analysis ($P < 0.05$, compared to isotype control group using Chi-square test). **ii)** viral RNA was detected in trachea, bronchus in all 3 isotype-control-treated animals and in the lung lobes of 2 of these animals (euthanized on 6 and 7 DPI), while none of the tissues tested displayed any detectable viral load in 2 animals (euthanized on 6 and 7 DPI) treated with 50mg/kg P4A1-2A. **iii)** clear alleviation of SARS-CoV-2 induced lung pathological changes in the 50 mg/kg P4A1-2A group compared to isotype control group. **iv)** dose dependency of anti-viral efficacy by P4A1-2A treatment, 50 mg/kg induced better viral clearance and alleviation of SARS-CoV-2 induced lung pathological changes than 10 mg/kg.

We rewrote the discussion section according to the reviewers' suggestion.

2. Why did the authors not assess BALS fluid which is the best indicator of viral replication in the upper respiratory tract of nonhuman primates and is commonly assessed in most studies?

Thank you for the comment. BALS fluid was not collected in the current study to avoid interference of viral spreading/clearance that can potentially result from the invasive procedure. Viral replications and spread were monitored by quantifying viral load in oropharyngeal swabs; nasal & rectal swabs (only nasal/rectal swabs on a few timepoints are positive, data not shown) and blood (all negative, data not shown) on different d.p.i., as well as viral load in tissues collected at necropsies at 5, 6, and 7 d.p.i..

Minor Comment

1. Why were the safety and PK tests done in cynomolgus monkeys and the efficacy tests done in rhesus monkeys?

Thank you for raising this point. In fact, we performed the PK and efficacy tests in two different laboratories. Cynomolgus monkey is the animal species commonly used for antibody safety evaluation and PK analysis and was used in our study because of relevance and wealth of historical data to compare to. Rhesus monkeys were selected for efficacy study because such a model has previously demonstrated viral replication/spreading and pathological changes following SARS-CoV-2 infection and is used for evaluating the efficacy of vaccines and neutralizing antibodies for COVID-19.

Reviewer #3:

The paper needs to be edited for better readability (english grammar) - see some suggestions below - but they are really all over the places.

Some references appear to be missing - with the vast number of publications on this subject, the authors could do a better job including all the work (similar to their) that has already been published.

Thank you for this comment. We have re-phrased the text and cite the literature. Relevant references were added throughout the manuscript to improve the quality. English grammar has been further revised.

The method used to identify the antibodies is not really “high-throughput single-cell B-cell receptor sequencing approach” since it involves using a “probe” to isolate the antibodies first and then sequence while what they mention could refer to just sequencing the B cell receptor. Maybe the authors can clarify this.

Yes, 'high-throughput single-cell B-cell receptor sequencing approach' refers to the sequencing of the B cell receptor of large number of B cells. We delete "high-throughput" in the revision to better clarify this description.

The statement (line 134) "This represents one of the most extensive coverages of residues involved in the Spike hACE2 interaction to date and may explain potent neutralization activity for P4A1." can be better explained in the context of the other similar antibodies (VH3-53) that have been isolated and bind the same site (from fig S4). It seems unclear if all the antibodies of that "class" have the same potency.

Thank you for this constructive suggestion. A sentence was added in reference of VH3-53 in general.

As such the title of the paper is misleading - it does not appear that this particular antibody has more exceptional binding coverage compared to the other published Mabs that bind the same site?

Thank you for this insightful comment, the word 'exceptional' was originally used to refer to the extensive coverage of residues involved in Spike-ACE2 binding (15 out of 17 with the 16th on the cutoff margin) and a relative bigger buried surface area (BSA). As suggested, the word 'exceptional' is changed to 'extensive'. The manuscript title has been revised to "A SARS-CoV-2 neutralizing antibody with extensive spike binding coverage modified for optimal therapeutic outcomes."

Regarding the structural analysis, the authors could add a PISA style table to include the interface.

Also it seems that the geometry of the NAG in the structure is wrong (from the PDB validation).

Thank you for this suggestion, we double check the PISA analysis and update the result; PISA style table is added to the supplementary data as Table S3 as suggested. The geometry of the NAG in the structure was also re-examined. As shown below, the NAG in our structure model (PDB code: 7CJF) fits the electron density (2fo-fc, contoured at 1 rmsd) well (upper panel), and is consistent with other released SARS-CoV-2 RBD structures (PDB code: 6M0J and 7KFY, lower panel).

It would have been interesting to compare PA41 IgG1 to the engineered PA41-IgG4 in monkeys as well as test the prophylactic role of the antibody.

Thank you, the authors agree with the notion. Extended half-life was demonstrated with engineered P4A1-IgG4 by comparison with historical IgG1 PK data. The other IgG1 versus engineered IgG4 differences are more difficult to demonstrate in monkeys as we were unable to identify monkey models that demonstrate SARS-CoV-2 ADE or other relevant properties.

The therapeutic setting (post viral challenge treatment) was chosen for this efficacy study because our initial clinical trials are intended for the treatment of mild-moderate COVID-19 infection (NCT04590430). In addition, therapeutic setting represents a more challenging system to demonstrate efficacy than a prophylactic setting. Others have shown that prophylactic treatment at lower doses results in better efficacy in term of viral load / spreading / pathological changes (Zost et al. *Nature*. 2020. 584: 443–449; Shi, R. et al. *Nature*. 2020. 584:120-124; Wang et al. *Nat Commun*. 2020. 11:5752) than higher doses in the therapeutic setting. This makes, in our opinion, prophylactic setting being less essential for P4A1-2A, that has demonstrated anti-viral efficacy in the therapeutic setting. Therefore, prophylactic setting was not tested in adherence to the

AAALAC 3R principle ('Reduce' in this case).

Thus again the title seems misleading as it is unclear if indeed the antibody therapeutic potential was optimized since there is no comparison to the IgG1 (the non optimized?).

Thank you for this insightful suggestion, the word 'Optimized' refers to the introduction of the properties (including longer half-life, lower ADE risks) that we believe are better suited for COVID-19 treatment. Whether the goal is achieved remains to be demonstrated in the clinic. The word is changed to 'modified for' to avoid confusion.

Examples where english writing could be improved and references added

The authors appreciate the careful review and editorial changes were made and references were added throughout the manuscript. Specially for the points below:

line 75 - add references
References were added.

line 83 change "also" to "was"
The word was corrected.

line 84: and raised a concern - rewrite that sentence..
The sentence was reworded.

line 86 - mutations rather than mutation sites.
The sentence was reworded.

something seems to be missing in the last sentence of the introduction
The sentence was reworded.

line 96 - reads weird
SARS-CoV-2 S protein binding B cells from PBMCs were enriched 97 with biotinylated S protein conjugated magnetic beads as probes as described
The sentence was reworded.

line 102 ND50 values of 5.212, 43.79, and 29.03 nM, shouldn't this be in nM/mL?
nM (nmol/L) appears to be correct.

line 105 - could add more references
More reference were added.

line 128 - P4A1 epitope
The word was added.

line 135- change broad coverage - here the authors likely mean the large overlap between ACE2 and P401 epitope on the RBD...?

The sentence was reworded.

line 137 - is there a reference?

The sentence was revised.

line 139 - missing “and thus are expected to..”

The words were added.

line 140 rewrite (it was supported by binding experiments...maybe add the mutations

The sentence was reworded.

line 176 - word seems to be missing

The word ‘mild’ was added.

Fig S4 - superposition...

The sentence was reworded as “Superposition of P4A1 (red, PDB 7CJF), CC12.1 (green, PDB 6XC2)...”

REVIEWER COMMENTS

Reviewer #2 (Remarks to the Author):

The authors cite a number of COVID-19 papers that employed rhesus macaques. The initial work with SARS-CoV-2 by most groups was done indeed done rhesus. However, subsequent work clearly shows that African greens are a much better model of disease. This should at least be addressed in the Discussion. While the authors attempt to address the question about statistics as noted before the group sizes were only $n = 3$ per group and because one animal per group was euthanized on days 5, 6, and 7, respectively it is only $n = 1$ per group when comparing parameters like tissue viral load or pathology.

Reviewer #3 (Remarks to the Author):

The authors have addressed most of the comments.

In light of the new SA strain (501.V2 lineage (also called B.1.351)), it could be worth to add binding of their antibody to RBD mutants including K417N and E484K (maybe also test binding to all 3 RBD mutations in that strain K417N and E484K and N501Y), especially since their antibody contacts K417N (and N501Y although P4A1 binding does not seem to be affected by that mutation).

This reviewer understands that this may increase the time to publication but it may be highly relevant as one concern is that this particular strain may escape the current vaccines.

Dear Reviewers,

The manuscript has been revised according to the reviewers' comments and formatted according to instructions.

Here are our point-to-point response to reviewers' comments (in blue)

Reviewer #2:

The authors cite a number of COVID-19 papers that employed rhesus macaques. The initial work with SARS-CoV-2 by most groups was done indeed done rhesus. However, subsequent work clearly shows that African greens are a much better model of disease. This should at least be addressed in the Discussion. While the authors attempt to address the question about statistics as noted before the group sizes were only $n = 3$ per group and because one animal per group was euthanized on days 5, 6, and 7, respectively it is only $n = 1$ per group when comparing parameters like tissue viral load or pathology.

Response:

We thank the reviewer for this suggestion. The author updated our reference accordingly, and we also notice that some research groups declared that AGMs would more accurately reflect human COVID-19 cases than other nonhuman primate species, however, the authors do not feel we have enough experience and information to make direct comparison among different NHP models.

To address reviewer's concern, we acknowledge other NHP models and their contributors by adding the references where NHP models were mentioned in the result section, and add the following sentences in the discussion section: "In addition to rhesus macaque model reported in this work, more NHP COVID-19 animal models, such as those established in African green monkey, cynomolgus macaques et.al, have also been developed to reflect clinical symptoms and underlying etiopathogenesis similar to those in COVID-19 patients. These disease relevant models provide alternative possibilities if we are to explore additional prophylactic application for P4A1-2A. Ultimately, the anti-viral efficacy of the antibody remains to be tested in clinical setting.

Reviewer #3:

The authors have addressed most of the comments.

In light of the new SA strain (501.V2 lineage (also called B.1.351)), it could be worth to add binding of their antibody to RBD mutants including K417N and E484K (maybe also test binding to all 3 RBD mutations in that strain K417N and E484K and N501Y), especially since their antibody contacts K417N (and N501Y although P4A1 binding does not seem to be affected by that mutation).

This reviewer understands that this may increase the time to publication but it may be highly relevant as one concern is that this particular strain may escape the current vaccines.

Response:

We thank the reviewer for this excellent suggestion and agree that the emergence of new clinical variants warrants an investigation into the activity of our antibody against these mutants. As an effort to continue testing our antibody against new available mutant, we have just completed the testing of binding to K417N, K444Q, V445A, L455F, E484K, F486V, Q493R and Q493K. The binding was not affected by E484K or other mutations, however, was reduced by K417N mutant. The information was added to the revised manuscript in Results and Supplementary Figure 5, and we will continue to test the antibody against emerging mutants.